# Improved Method for Dental Pulp Stem Cell Preservation and Its Underlying Cell Biological Mechanism

**DOI:** 10.3390/cells12172138

**Published:** 2023-08-24

**Authors:** Mai Takeshita-Umehara, Reiko Tokuyama-Toda, Yusuke Takebe, Chika Terada-Ito, Susumu Tadokoro, Akemi Inoue, Kohei Ijichi, Toshio Yudo, Kazuhito Satomura

**Affiliations:** Department of Oral Medicine and Stomatology, School of Dental Medicine, Tsurumi University, 2-1-3 Tsurumi, Tsurumi-ku, Yokohama 230-8501, Kanagawa, Japan; umemai.0202@gmail.com (M.T.-U.); takebe-yusuke@tsurumi-u.ac.jp (Y.T.); terada-chika@tsurumi-u.ac.jp (C.T.-I.); tadokorosusumu@yahoo.co.jp (S.T.); i-ake@mvb.biglobe.ne.jp (A.I.); nitou2284@icloud.com (K.I.); ut2015mail27integral@gmail.com (T.Y.); satomura-k@tsurumi-u.ac.jp (K.S.)

**Keywords:** dental pulp stem cell, regenerative medicine, preservation method

## Abstract

Dental pulp stem cells (DPSCs) are considered a valuable cell source for regenerative medicine because of their high proliferative potential, multipotency, and availability. We established a new cryopreservation method (NCM) for collecting DPSCs, in which the tissue itself is cryopreserved and DPSCs are collected after thawing. We improved the NCM and developed a new method for collecting and preserving DPSCs more efficiently. Dental pulp tissue was collected from an extracted tooth, divided into two pieces, sandwiched from above and below using cell culture inserts, and cultured. As a result, the cells in the pulp tissue migrated vertically over time and localized near the upper and lower membranes over 2–3 days. With regard to the underlying molecular mechanism, SDF1 was predominantly involved in cell migration. This improved method is valuable and enables the more efficient collection and reliable preservation of DPSCs. It has the potential to procure a large number of DPSCs stably.

## 1. Introduction

Regenerative medicine requires three components: cells, scaffolds, and bioactive molecules (growth factors) [1,2,3,4,5,6]. It is particularly important to secure high-quality stem cells efficiently and reliably. Adult stem cells (AS cells), embryonic stem cells, and induced pluripotent stem cells (iPS cells) represent typical cell sources for regenerative medicine [7,8,9,10,11,12,13,14]. Of these, AS cells are significant because they can overcome problems, such as tumorigenesis [8,15,16] and immunological rejection [8,17,18]. In 2000, Gronthos and colleagues identified MSCs from the dental pulp of permanent teeth, which were designated dental pulp stem cells (DPSCs) [19]. In that study, the dental pulp was collected from the extracted tooth, and the migrated cells were cultured to establish dental pulp stem cells. DPSCs, like other AS cells, have high proliferative potential and multipotency [1,19,20,21,22]. They are also more readily available than other AS cell types [21,23,24]. Therefore, DPSCs are regarded as a high-quality source of regenerative medicine [21,23,25,26].

The current method for collecting and preserving DPSCs is to collect pulp tissue from extracted teeth, culture the growing cells until they reach subconfluency, and cryopreserve them [23,27,28,29]. This method has a variety of problems, however, including the need for a long culture period before cell collection [24,30], a high risk of contamination [31,32,33,34], high cost [34], and the need to secure storage space for cryopreservation [23,35]. Therefore, it is necessary to establish a low-cost, simple, and reliable cell collection system to procure and stably secure more DPSCs for the efficient implementation of regenerative medicine [27,29]. We established a new cryopreservation method that can effectively collect and preserve human DPSCs [27]. For this method, pulp tissue is collected from extracted teeth, shredded into tissue pieces, placed into a culture disk with medium for 5 days, and cryopreserved. After thawing the tissue pieces, the proliferation capacity of the cells obtained by a migration method was comparable to that of cells obtained by conventional methods. Moreover, the expression pattern of the surface markers of the cells obtained was similar to that of cells obtained by conventional methods. Furthermore, we confirmed that the cells obtained by this method could differentiate into osteoblasts, adipocytes, and chondrocytes. These results indicate that the cells obtained by our method have the characteristics of mesenchymal stem cells, similar to those obtained by conventional methods. Using this method, pulp tissue may be transported to a facility equipped with a cell preservation system and immersed into a culture medium for 5 days. The cells with stem cell characteristics can be readily collected from pulp tissue reliably and inexpensively. However, because the actual transportation period requires 2–3 days, it is necessary to establish an improved method to collect cells over a shorter period. In this study, we hypothesized that an improved method would involve sandwiching pulp tissue from above and below, rather than placing it on a disc, so that DPSCs may be collected more efficiently.

## 2. Materials and Methods

### 2.1. Collection of Dental Pulp Tissues

This study was approved by the Research Ethics Review Committee at Tsurumi University School of Dental Medicine (approval number: 121011). Dental pulp tissues were obtained from the third molars of healthy patients (20–34 years of age, 38 teeth of 35 patients) extracted for clinical reasons at Tsurumi University Dental Hospital. Before extraction, informed written consent was obtained from all patients. The third molars used in this study had no caries or restorative intervention. Extracted teeth were cut at the cement-enamel junction using a diamond bur and dental pulp tissues were gently harvested under sterile conditions.

### 2.2. Improved Culture Method for Dental Pulp Tissues

Dental pulp tissue collected from the crown of the tooth was divided into two pieces. Each piece was sandwiched from above and below using Falcon^®^ cell culture inserts (the pore size is 0.4 µm, Falcon, Corning, NY, USA) so that the tissue was in contact with both insert membranes, and placed into a 6-well multiplate (Falcon), and cultured in 4.0 mL of MEMα (Wako Pure Chemical Co., Ltd., Osaka, Japan) supplemented with 10% fetal bovine serum (Biological Industries, Kibbutz Beit-Haemek, Israel), 100 U/mL penicillin, and 100 µg/mL streptomycin (Invitrogen, Carlsbad, CA, USA) at 37 °C in a humidified atmosphere of 5% CO_2_ in air for 5 days. This method was designated the “Improved culture method for dental pulp tissue” (Improved CMDPT) (Figure 1). Note that no contamination was observed in the primary culture.

### 2.3. Histological Examination of Dental Pulp Tissue Pieces Cultured Using the Improved CMDPT

To identify the cell dynamics in dental pulp tissues cultured using the Improved CMDPT, dental pulp tissues were fixed every 24 h with 4% paraformaldehyde phosphate buffer solution (PFA; Wako Pure Chemical Industries Ltd., Osaka, Japan). At this time, the tissues were taken out while sandwiched between the insert membranes and fixed together with the insert membranes. They were embedded in paraffin and sectioned at 5-µm thickness. The paraffin sections were deparaffinized, hydrated, and stained with hematoxylin and eosin (H&E) to observe the cell dynamics of the dental pulp tissues. To confirm the presence of cells that grew from the dental pulp tissue onto the upper and lower membranes, the membranes were separately removed from the dental pulp tissue and fixed with 1% glutaraldehyde in phosphate-buffered saline (PBS, pH 7.4) for 15 min. The membranes were washed with saline and stained with 0.02% crystal violet solution overnight to identify cells that migrated out from dental pulp tissue.

### 2.4. Comparison with the NCM Method (Previous Study)

We compared the time required to obtain DPSCs by the NCM method [27], which was performed in the previous study, and the Improved CMDPT method, which was performed in this study. First, Improved CMDPT was performed as described above, and the number of cells migrating to the vicinity of the upper and lower membranes was counted. In addition, the NCM method, which was used in a previous study, was performed. Therefore, a piece of dental pulp tissue was placed on one cell insert, and the number of cells migrating to the vicinity of the membrane after 120 h reported in the previous study was counted and compared.

### 2.5. Immunohistochemical Examination

An immunohistochemical examination was carried out to analyze the biological characteristics of the cells in the dental pulp tissues cultured using the improved CMDPT. The sections were deparaffinized with xylene, rehydrated in descending concentrations of ethanol, and washed in PBS. After antigen retrieval in a 0.01 mol/L sodium-citrate-buffered solution (pH 6.0) at 98 °C for 45 min, the sections were cooled and washed well with PBS. Endogenous peroxidase was blocked by treatment with 3% H_2_O_2_ in methanol for 1 h at room temperature (RT). After treatment with 10% normal goat serum at RT for 10 min, the sections were incubated with rabbit anti-Ki-67 antibody (Abcam, Cambridge, UK) (diluted 1:200 in PBS containing 1% bovine serum albumin) or rabbit anti-stromal cell-derived factor-1(SDF1) antibody (Abcam, Cambridge, UK) (diluted 1:1000 in PBS containing 1% bovine serum albumin) at 4 °C overnight. After washing with PBS, the localization of each molecule was determined using a Histofine SAB-PO (R) kit (Nichirei Biosciences, Inc., Tokyo, Japan) and 3, 3′-diaminobenzidine (DAB) substrate kit (Nichirei Biosciences, Inc.). The sections were counterstained with hematoxylin and mounted. The specificity of the immunoreaction was confirmed by incubation with normal rabbit IgG instead of primary antibody.

### 2.6. Cell Migration Assay

To confirm the role of SDF1 in cell dynamics in dental pulp tissue, we performed a migration assay in which recombinant SDF1 protein was added to the culture of DPSCs. Based on the manufacturer’s instructions, Falcon^®^ cell culture inserts (8 µm pore size; Falcon-cat353093) were placed into 6-well Transwell plates. Two mL of cell suspension containing 5.0 × 10^5^ cells/mL of DPSCs in serum-free medium was added to the inside of the inserts and 1 mL of serum-free medium was added to the bottom of the Transwell plate. Recombinant SDF1 protein (Sigma-Aldrich Co, St. Louis, MO, USA) or normal goat IgG (MBL Life Science, Nagoya, Japan) was added to the inside of the inserts at a concentration of 100 ng/mL and cultured for 24 h at 37 °C in a humidified atmosphere of 5% CO_2_ in air. The tip of the cotton swab was hydrated with water and pressed against a clean hard surface to flatten the end of the swab. The medium was carefully aspirated from the inside of the insert, which was gently wiped with the cotton swab to remove non-invasive cells. The insert was carefully washed twice with PBS and the cells that had passed through the insert and reached the underside were fixed with 1% glutaraldehyde for 15 min, washed with PBS, and stained with 0.02% crystal violet solution for 1 h. Cell numbers were counted under a microscope and the effect of recombinant SDF1 protein on the migration of DPSCs was examined compared with that of the control group.

### 2.7. Effects of SDF1 Inhibition on the Migration of DPSCs Cultured Using the Improved CMDPT

To confirm that SDF1 is involved in the migration of DPSCs cultured using the Improved CMDPT, the effect of anti-SDF1 neutralizing antibody on the migration of DPSCs was determined. Improved CMDPT was performed in the same manner as previously described. Therefore, the dental pulp tissue was divided into two pieces. One piece was treated with 6.0 µg/mL anti-SDF1 neutralizing antibody (R&D Systems, Minneapolis, MN, USA) diluted in PBS and the other was treated with 6.0 µg/mL normal goat IgG (MBL) diluted in PBS as a control. Dental pulp tissues were cultured for 48, 72, and 96 h. Tissue sections were prepared as described above and stained with H&E. The effect of the anti-SDF1 neutralizing antibody was confirmed by histological observation.

### 2.8. Statistical Analysis

The results are presented as the mean ± standard errors. All data are representative of at least three individual experiments and analyzed with the Mann–Whitney U-test. Differences were considered significant at *p* < 0.05.

## 3. Results

### 3.1. Histological Examination of Dental Pulp Tissue Cultured by the Improved CMDPT

Histological examination of dental pulp tissue cultured by the Improved CMDPT revealed that cell localization in the tissue changed over time (Figure 2a–e). After 72 h of culture, the cells became localized near the upper and lower membranes, and almost no cells were observed in the central area after 96 h (Figure 2d,e). To determine whether cell localization was the result of cell proliferation or migration, we examined the expression of Ki-67, a marker of cell proliferation. No Ki-67 expression was observed in the cells of the dental pulp tissue at any time point (Figure 2f–j). This suggests that the localization change of DPSCs was not the result of cell proliferation, but due to the vertical migration of the cells toward the upper and lower membranes. After 96 h of culture, the DPSCs migrated from the dental pulp tissue onto the membrane contacting the dental pulp tissue (Figure 2k–v). This indicates that the DPSCs actively migrated toward the membranes under the Improved CMDPT.

### 3.2. Comparison of the Number of Cells Migrating to the Vicinity of the Membrane

We compared the number of cells migrating to the vicinity of the membrane between the improved CMDPT of this time and the method in the previous study. As a result, in Improved CMDPT, it was found that the number of cells that migrated to the vicinity of the membrane in 48 h was comparable to the number of cells observed in 120 h in the previous study (Figure 3). Furthermore, at 72 h of Improved CMDPT, it was found that a greater number of cells migrated to the vicinity of the membrane than were observed in previous studies. (Figure 3). From these results, it was found that cells migrate to the vicinity of the membrane early in Improved CMDPT. In addition, it is suggested that DPSCs can be obtained more efficiently in the Improved CMDPT because the moving distance is shorter in the upward and downward direction than in the one direction.

The number of cells migrated to the vicinity of the membrane in 48 h of Improved CMDPT was comparable to the number of cells observed in 120 h with NCM. In addition, at 72 h of Improved CMDPT, it was found that a greater number of cells migrated to the vicinity of the membrane than had been observed in the previous study.

### 3.3. Identification of Factors Involved in Cell Migration

Next, we examined the expression of SDF1, which is a chemotactic factor for mesenchymal stem cells, by immunohistochemistry to identify the factors affecting cell migration in dental pulp tissue. The expression of SDF1 was observed in the central portion of the dental pulp tissue at 72 h of culture when active cell migration was observed (Figure 3a,c). In contrast, no SDF1 expression was evident at 96 h of culture, when almost all the cells had migrated vertically toward the membranes (Figure 4b,d). These findings suggest that SDF1 is involved in DPSC migration in the dental pulp tissue under the Improved CMDPT.

### 3.4. Effect of SDF1 on the Migration of DPSCs In Vitro

Using a cell migration assay, we determined whether SDF1 exhibited a migration-promoting effect on DPSCs in vitro. The results indicated a significant cell migration-promoting effect of SDF1 in the DPSC cultures (Figure 5).

### 3.5. Effect of SDF1neutralizing Antibody on the Migration of DPSCs under the Improved CMDPT

We determined whether SDF1 inhibition affected the migration of DPSCs cultured by the Improved CMDPT. In the culture without SDF1-neutralizing antibody, DPSCs were localized near the membranes after 48 h (Figure 6a). In contrast, no cell migration was observed in the presence of SDF1-neutralizing antibody (Figure 6d). After 72 h of culture, DPSCs continued to migrate toward the membranes (Figure 6b), whereas no migration was observed in the presence of SDF1-neutralizing antibody (Figure 6e). After 96 h of culture, the cells in the control group, which were localized in the vicinity of the membranes after 72 h of culture, were nearly absent, presumably because of growth outside the dental pulp tissue (Figure 6c). In contrast, in the group treated with SDF1-neutralizing antibody, some DPSCs were localized near the upper and lower membranes (Figure 6f). These results indicate that DPSC migration in the dental pulp tissue cultured by the Improved CMDPT was inhibited by SDF1.

## 4. Discussion

DPSCs are one of the readily available adult mesenchymal stem cells and are considered a potential cell source for regenerative medicine because of their good proliferative and multipotent characteristics [22,36,37,38,39,40,41]. DPSCs can differentiate into other cell types (all three germ layers), including odontogenic, osteogenic, chondrogenic, neurogenic, myogenic, and endothelial cell lines as well as insulin-producing cells [36,37,41,42]. The broad spectrum of differentiation potential and relative ease of collection make DPSCs a valuable cell source for therapy and regenerative medicine. Recent in vitro studies indicate their potential use for the treatment of endocrine disorders (diabetes mellitus), neurodegenerative diseases (Parkinson’s disease, Alzheimer’s disease, stroke), spinal injury, peripheral nerve injury, and the repair and regeneration of bones and cartilage (osteoporosis, osteoarthritis) [23,43,44,45,46,47]. Therefore, DPSCs may be useful as a cell source for regenerative medicine; however, to obtain sufficient DPSCs for regenerative medicine in the future, it is essential to establish a robust technique for DSPC collection and preservation without impairing their stem cell properties [48,49].

Previously, we established a new method of adhering dental pulp tissue to a dish surface, culturing it for several days, and cryopreserving it successfully [27]. In the present study, we developed an improved culture method for dental pulp tissue that is more efficient for collecting and cryopreserving DPSCs. In the previous method, we confirmed that cells were localized to the peripheral part of the dental pulp tissue contacting the dish surface and these cells were successfully cryopreserved. In the present study, we hypothesized that sandwiching the dental pulp tissue with membranes from above and below would increase the efficiency of cell collection and ensure their stability during transport. As a result, we confirmed that cells in the dental pulp tissue migrated over time toward the two membranes after 48 h of culture. Moreover, it was found that the improved method not only migrated cells earlier but also increased the number of migrated cells compared to the method in the previous study. Compared to the distance required for movement in one direction so far, the total distance cells migrate is reduced because it can move both upward and downward direction. As a result, it is thought that this leads to efficient collection of cells. This indicates that DPSCs can be collected more efficiently and the culture period may be reduced compared with that of the previous method. We further demonstrated that the DPSCs collected by this method maintain both proliferative and multipotent potential (shown in Appendix A). These results indicate that the new method is more efficient for collecting DPSCs from dental pulp tissue in terms of collection efficiency and time reduction.

Next, we examined the cell dynamics in the dental pulp tissue subjected to the Improved CMDPT. We initially focused on SDF1, a member of the CXC chemokine subfamily [50], which promotes the recruitment of MSCs and inflammatory cells to injured sites and regulates the repair of tissue and organ damage. Previous studies demonstrated that DPSCs also express SDF1 and its receptor CXC chemokine receptor 4 (CXCL4). SDF1 exerts a chemotactic effect on DPSCs by inducing directional migration [51]. An immunohistochemical examination of dental pulp tissue cultured by the Improved CMDPT revealed SDF1 expression when DPSCs started to migrate. In contrast, no SDF1 expression was observed in dental pulp tissue once cell migration was completed. These results suggest that SDF1 is significantly involved in DPSC migration in this culture method. In addition, the Recombinant SDF1 protein promoted the migration of DPSCs, whereas an SDF1-neutralizing antibody inhibited DPSC migration under the Improved CMDPT. Based on these results, we confirmed that SDF1 is significantly involved in the mechanism of cell migration in the Improved CMDPT. Interestingly, DPSC migration was not completely inhibited but delayed by SDF1 neutralizing antibody. These results suggested that the mechanism of DPSC migration in the Improved CMDPT is influenced not only by SDF1 but also by other factors. We are planning to examine the precise mechanism of DPSC migration under the Improved CMDPT by identifying these additional factors. Currently, as shown in Appendix A, we are conducting a comprehensive multiplex analysis of whole dental pulp tissue processed by the Improved CMDPT to identify candidate factors.

Previously, we established a novel cryopreservation method that enables the efficient collection of DPSCs [27]. By immersing dental pulp tissue in a culture medium, DPSCs could be cultured even during transport, thus effectively utilizing this time to transport the DPSCs to a specialized facility where they could be stored and lowering the overall cost of the procedure. The newly developed Improved CMDPT reduces the culture period to 2–3 days, which is almost the same as the transportation period to the cell storage facility, making it more efficient. Considering the fact that the current method of collecting stem cells from dental pulp tissue requires long-term culture, high cost, and an increased risk of contamination [23], the techniques for securing DPSCs more stably and efficiently are essential for regenerative medicine [27,29].

The Improved CMDPT established in this study represents a simple and efficient method to collect DPSCs and overcomes the aforementioned problems in the process of DPSC isolation, culture, and preservation. We believe that this method will significantly contribute to securing cell sources for regenerative medicine. We will use this Improved CMDPT to transport dental pulp tissue, collect DPSCs, and confirm the possibility of using them for regenerative medicine. We also will proceed with their clinical application in a multi-institutional collaboration to determine whether the Improved CMDPT is suitable as a cell collection method for DPSC banks.

## 5. Conclusions

The Improved CMDPT is a useful method for the efficient collection and reliable preservation of DPSCs. This method has the potential to secure a large number of DPSCs stably. We plan to further elucidate the mechanism underlying this method and embark on clinical studies.

## Figures and Tables

**Figure 1 cells-12-02138-f001:**
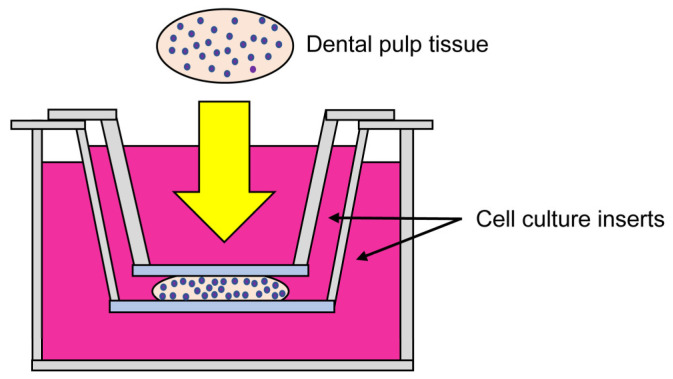
Schematic representation of the Improved CMDPT. The collected dental pulp tissue was sandwiched (yellow arrow) from above and below using Falcon^®^ cell culture inserts and cultured at 37 °C in a humidified atmosphere of 5% CO_2_ in air.

**Figure 2 cells-12-02138-f002:**
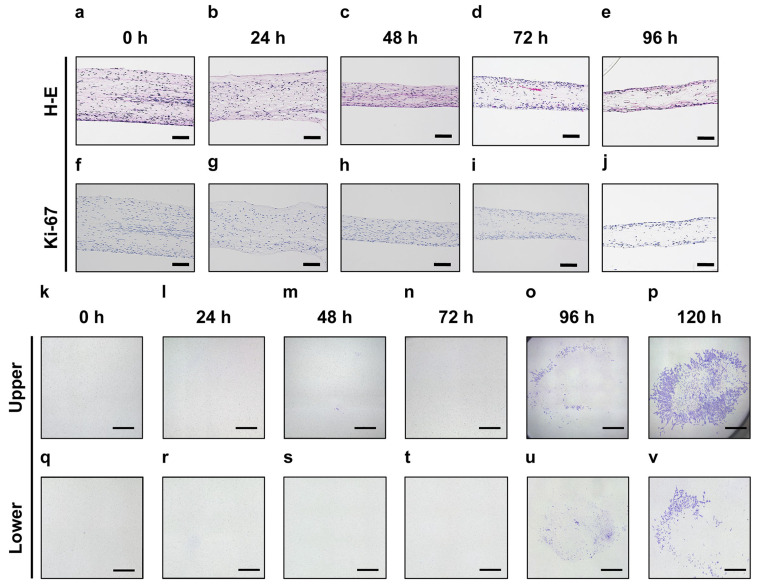
Histological findings of dental pulp tissue cultured using the Improved CMDPT. Histological images at 0 h (**a**,**f**,**k**,**q**), 24 h (**b**,**g**,**l**,**r**), 48 h (**c**,**h**,**m**,**s**), 72 h (**d**,**i**,**n**,**t**), 96 h (**e**,**j**,**o**,**u**), and 120 h (**p**,**v**) of Improved CMDPT. (**a**–**e**) H&E staining. After 72 h of culture, the cells were localized near the upper and lower membranes, whereas almost no cells were observed in the central area after 96 h. (**f**–**j**) Immunohistochemical staining for Ki-67. No expression of Ki-67 was observed in the sections. (**k**–**p**) Crystal violet staining of the upper membrane. (**q**–**v**) Crystal violet staining of the lower membrane. After 96 h of culture, the cells had migrated from the dental pulp tissue onto the membrane contacting the dental pulp tissue. (**a**–**j**) Scale bars are 100 µm; (**k**–**v**) scale bars are 500 µm.

**Figure 3 cells-12-02138-f003:**
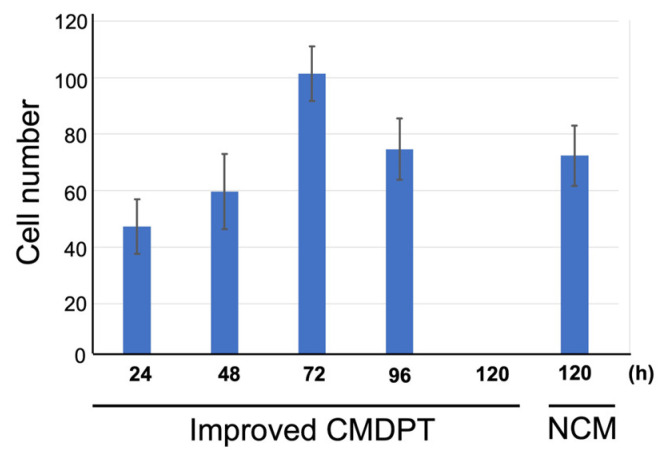
Comparison of the number of cells migrating to the vicinity of the membrane.

**Figure 4 cells-12-02138-f004:**
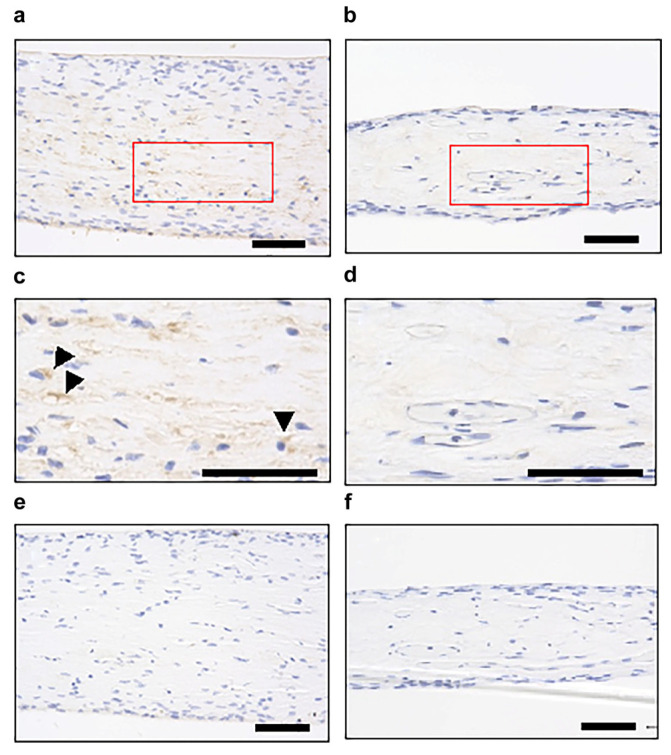
Immunohistochemical localization of SDF1 in dental pulp tissue cultured using the Improved CMDPT. (**a**–**d**) Immunohistochemical staining for SDF1 ((**c**,**d**) are high magnification images of the red box area, respectively), (**e**,**f**) Negative control. (**a**,**c**,**e**) After 72 h of culture, (**b**,**d**,**f**). After 96 h of culture. SDF1 expression was observed in the center portion of the dental pulp tissue after 72 h of culture when active cell migration was observed. In contrast, no expression of SDF1 was observed at 96 h of culture, when almost all of the cells had migrated vertically toward the membrane. Arrowheads indicate the localization of SDF1. (**a**–**f**) Scale bars are 50 µm.

**Figure 5 cells-12-02138-f005:**
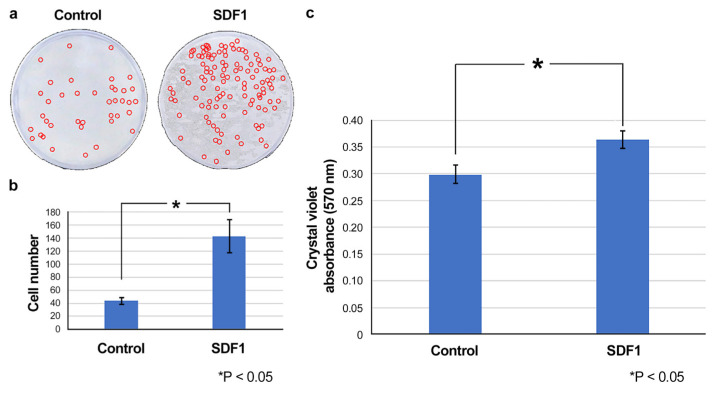
Effect of SDF1 on the migration of DPSCs in vitro (**a**) Photo of the culture insert. (**b**) The number of migrating cells. (**c**) The absorbance of crystal violet at 570 nm. A significant cell-migration-promoting effect of SDF1 was observed in the culture of DPSCs.

**Figure 6 cells-12-02138-f006:**
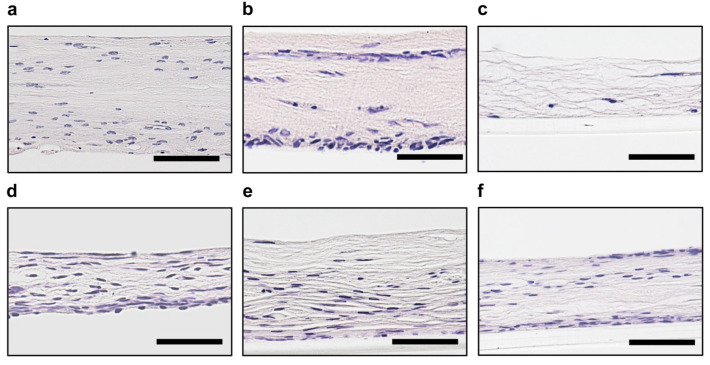
Effect of SDF1-neutralizing antibody on the migration of DPSCs under Improved CMDPT. Histological images after 48 h (**a**,**d**), 72 h (**b**,**e**), 96 h (**c**,**f**) of cultures using the Improved CMDPT. (**a**–**f**) H&E staining. (**a**–**c**) Control. (**d**–**f**) SDF1-neutralizing antibody was added. After 48 h of culture, DPSCs were localized near the membrane in the culture without an SDF1-neutralizing antibody. In contrast, no cell migration was observed in the presence of SDF1-neutralizing antibody. After 72 h of culture, DPSCs migrated toward the membranes, whereas no migration was observed in the cultures containing SDF1-neutralizing antibody. After 96 h of culture, the cells in the control group, which had been localized in the vicinity of the membrane after 72 h of culture, were almost absent, presumably because of growth outside the dental pulp tissue. In contrast, in the SDF1-neutralizing-antibody-treated group, some DPSCs were localized near the upper and lower membranes. (**a**–**f**) Scale bars are 50 µm.

## Data Availability

The data presented in this study are available in the Appendix A.

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
