# Peer review of "Improved Method for Dental Pulp Stem Cell Preservation and Its Underlying Cell Biological Mechanism"

_cells, 2023, doi:10.3390/cells12172138_

Round 1

Reviewer 1 Report

The authors have submitted an interesting paper on the isolation of stem cells from pulp tissue. The paper is well written and the content might be practically useful. However, the authors should address the following points before a decision is made, as some parts are either unclear or has major shortcomings.

The abstract is concise and clear;

For in introduction, please rephrase the following sentence: ''AS cells are significant because''; in the introduction, could you please explain which the standard/conventional procedure for obtaining stem cells from dental pulp?

For the methods, how many molars and from how many patients were extracted? How sterility was preserved during pulp tissue isolation?

Please specify the pore size of cell culture inserts? Was the tissue evenly touching the membranes? Which was the thickness of the tissue? Which volume of medium have you used in each 6 well plate.

For paragraph 2.6, were the tissues grown in transwells as before? Please better describe the method.

Figure. 2 Make the figures larger. Check the spaces between words: Scale bars are100 μm.

Section 2.3 is unclear. Were the tissues embedded with their inserts? Please, better explain this part.

Why no characterization of the extracted stem cells was done? FACS, immuno, ability to differentiate, purity. This is the main weakness of this work, together with lack of the indication of the number of repeats. 

The authors state that the this method is an improvement in comparison to their previous one. Can the author give data and stats to support their statement?

I think the English Language is fine and text is fluent.

Author Response

July 3, 2023

Dear Reviewer #1

cells

Re:   Manuscript ID: cells-2491615

Title: Improved method for dental pulp stem cell preservation and its underlying cell biological mechanism

Thank you for your valuable comments concerning our manuscript entitled " Improved method for dental pulp stem cell preservation and its underlying cell biological mechanism."

We have carefully studied your comments and made the necessary corrections, and are sending here the revised manuscript again. The corrected document has colored text.

Our responses to your comments are as follows:

Response to the comments of Reviewer #1

  1. For in introduction, please rephrase the following sentence: ''AS cells are significant because''; in the introduction, could you please explain which the standard/conventional procedure for obtaining stem cells from dental pulp?

Response

Thank you for your comment. I added the explanation in lines 34-36 as follows:

" In that study, the dental pulp was collected from the extracted tooth, and the migrated cells were cultured to establish dental pulp stem cells.” (L34 to L36 on page 1, Revision text)

  1. For the methods, how many molars and from how many patients were extracted? How sterility was preserved during pulp tissue isolation?

Response

Thank you for your comment.

In this study, 38 teeth were extracted from 35 patients and pulp was obtained from them. DPSCs could be obtained from 36 dental pulps by the improved CMDPT method. In a previous study, 30 teeth were extracted from 30 patients and pulp was obtained from them. DPSCs could be obtained by the NCM method from 28 dental pulps. The DPSCs obtained in each study showed similar cell dynamics behavior. From this, it was thought that DPSCs collected by the improved CMDPT method also have stem cell characteristics, similar to DPSCs obtained by the previous NCM method.

So, I added the explanation in line 68-69, “x teeth of x patients”. (L68 to L69 on page 2, Revision text)

Next, the dental pulp is normally completely covered with a strong tooth substance and exists in a sterile state. In this study, the subjects were non-carious teeth that had not yet erupted or were extracted due to malposition. Therefore, the possibility of infection was minimal, and in fact no contamination was observed in the primary culture.

So, I added the sentence in lines 83-84 as follows:

“Note that, no contamination was observed in the primary culture.” (L83 to L84 on page 2, Revision text)

  1. Please specify the pore size of cell culture inserts? Was the tissue evenly touching the membranes? Which was the thickness of the tissue? Which volume of medium have you used in each 6 well plate.

Response

Thank you for your comment. We clarified changes in method as follows:

“the pore size is 0.4 mm” (L76 to L77),

“so that the tissue was in contact with both insert membranes, and “ (L77 to L78),

“4.0 mL of” (L78 on page 2, Revision text).

And, dental pulp tissues having sizes of approximately 4.0 mm were divided and sandwiched between the upper and lower membranes, and the average distance between the membranes was 159.2±55.8 mm at 0 h and 122.8±23.0 mm at 24 h.

  1. For paragraph 2.6, were the tissues grown in transwells as before? Please better describe the method.

Response

Thank you for your comment. Yes, I cultured the tissue sandwiched between cell inserts as before. The only difference is the presence or absence of neutralizing antibodies. Based on your comment, I added the following:

“Improved CMDPT was performed in the same manner as previously described. Therefore, the dental pulp tissue was divided into two pieces.” (L151to L152 on page 4, Revision text).

  1. Figure. 2 Make the figures larger. Check the spaces between words: Scale bars are100 μm.

Response

Thank you for your comment. Space is wrong, it has changed. Thank you very much.

And, I had made the figures larger. Is it enough?

  1. Section 2.3 is unclear. Were the tissues embedded with their inserts? Please, better explain this part.

Response

Thank you for your comment. It seems my explanation was not clear. Added the following sentences:

“At this time, the tissues were taken out while sandwiched between the insert membranes and fixed together with the insert membranes.” (L93 to L95 on page 3, Revision text).

  1. Why no characterization of the extracted stem cells was done? FACS, immuno, ability to differentiate, purity. This is the main weakness of this work, together with lack of the indication of the number of repeats.

The authors state that the this method is an improvement in comparison to their previous one. Can the author give data and stats to support their statement?

Response

Thank you for your comment.

Thank you for your valuable suggestions for improvement of our manuscript. As per your comment, we have not repeated the examination for stem cell characterization. This is because, as I answered in comment 2, DPSCs could be collected in the same way compared to NCM, and the characters of the collected DPSCs did not change the cell dynamics and showed the same behavior. Moreover, we found that the cells were obtained earlier compared to previous methods. This is because in the course of our current study, we compared the number of cells that could be collected over time between the method using a single insert, which is the previous study, and the improved CMDPT method. Add the result to figure 3. As can be seen from this graph, a sufficient number of DPSCs can be obtained early compared to previous methods.

Therefore, according to your indication, I added “figure 3” and the following text.

“2.4. Comparison with the NCM method (previous study)

We compared the time required to obtain DPSCs by the NCM method [28], which was performed in the previous study, and the Improved CMDPT method, which was performed in this study. First, Improved CMDPT was performed as described above, and the number of cells migrating to the vicinity of the upper and lower membranes was counted. In addition, the NCM method, which was used in previous study, was performed. Therefore, a piece of dental pulp tissue was placed on one cell insert, and the number of cells migrating to the vicinity of the membrane after 120 hours reported in the previous study was counted and compared.” (L103 to L111 on page 3, Revision text).

“3.2. Comparison of the number of cells migrating to the vicinity of the membrane

We compared the number of cells migrating to the vicinity of the membrane between the improved CMDPT of this time and the method in the previous study. As a result, in Improved CMDPT, it was found that the number of cells migrated to the vicinity of the membrane in 48 hours was comparable to the number of cells observed in 120 hours in the previous study (Figure 3). Furthermore, at 72 hours of Improved CMDPT, it was found that the greatly number of cells migrated to the vicinity of the membrane than were observed in previous studies. (Figure 3). From these results, it was found that cells migrate to the vicinity of the membrane early in Improved CMDPT. In addition, it is suggested that DPSCs can be obtained more efficiently in the Improved CMDPT because the moving distance is shorter in the upward and downward direction than in the one direction.” (L187 to L196 on page 5, Revision text)

“Figure 3. Comparison of the number of cells migrating to the vicinity of the membrane

The number of cells migrated to the vicinity of the membrane in 48 hours of Improved CMDPT was comparable to the number of cells observed in 120 hours with NCM. In addition, at 72 hours of Improved CMDPT, it was found that the greatly number of cells migrated to the vicinity of the membrane than had been observed in the previous study.” (L198 to L202 on page 6, Revision text)

“Moreover, it was found that the improved method not only migrated cells earlier, but also increased the number of migrated cells compared to the method in the previous study. Compared to the distance required for movement in one direction so far, the total distance cells migrate is reduced because it can move both upward and downward direction. As a result, it is thought that this leads to efficient collection of cells.” (L280 to L285 on page 8-9, Revision text)

We believe the manuscript has been improved satisfactorily and hope that it is now acceptable for publication in cells

Yours sincerely,

Reiko Tokuyama-Toda, DDS, PhD 

Reviewer 2 Report

In this study, the authors improved their established cryopreservation method for collecting and preserving dental pulp stem cells (DPSCs). To test the efficiency of this improved method (CMDPT), the authors examine the dental pulp tissue cultured by this method and find that cells move vertically over a period of 2-3 days, congregating near the upper and lower membranes. They further demonstrate that SDF1 is the primary driver of this cell migration. Finally, the authors claim that this improved method (CMDPT) is valuable for regenerative medicine due to its high growth potential, multipotency, and easy accessibility. This improved method does provide a promising way for efficiently harvesting DPSCs for regenerative medicine. However, for robustness, I suggest that the authors introduce their earlier non-refined method as a control. This would provide a comparative baseline and allow a quantifiable evaluation of the improvement this new method brings. Moreover, I recommend that the discussion section be revisited. Currently, it tends to reiterate findings already covered in the results section. Instead, it would be more beneficial if the authors focused on contrasting their new method with traditional DPSC collection techniques and their previously established method.

Author Response

July 3, 2023

Dear Reviewer #2

cells

Re:   Manuscript ID: cells-2491615

Title: Improved method for dental pulp stem cell preservation and its underlying cell biological mechanism

Thank you for your valuable comments concerning our manuscript entitled " Improved method for dental pulp stem cell preservation and its underlying cell biological mechanism."

We have carefully studied your comments and made the necessary corrections, and are sending here the revised manuscript again. The corrected document has colored text.

Our responses to your comments are as follows:

Response to the comments of Reviewer #2

  1. In this study, the authors improved their established cryopreservation method for collecting and preserving dental pulp stem cells (DPSCs). To test the efficiency of this improved method (CMDPT), the authors examine the dental pulp tissue cultured by this method and find that cells move vertically over a period of 2-3 days, congregating near the upper and lower membranes. They further demonstrate that SDF1 is the primary driver of this cell migration. Finally, the authors claim that this improved method (CMDPT) is valuable for regenerative medicine due to its high growth potential, multipotency, and easy accessibility. This improved method does provide a promising way for efficiently harvesting DPSCs for regenerative medicine.

Response

Thank you very much. I am encouraged by your comment.

  1. However, for robustness, I suggest that the authors introduce their earlier non-refined method as a control. This would provide a comparative baseline and allow a quantifiable evaluation of the improvement this new method brings.

Response

Thank you very much for your valuable suggestions for improvement of our manuscript. In this study, we found that the cells were obtained earlier compared to previous methods. This is because in the course of our current study, we compared the number of cells that could be collected over time between the method using a single insert, which is the previous study, and the improved CMDPT method. Add the result to figure 3. As can be seen from this graph, a sufficient number of DPSCs can be obtained early compared to previous methods.

Therefore, according to your indication, I added figure 3 and the following text.

“2.4. Comparison with the NCM method (previous study)

We compared the time required to obtain DPSCs by the NCM method [28], which was performed in the previous study, and the Improved CMDPT method, which was performed in this study. First, Improved CMDPT was performed as described above, and the number of cells migrating to the vicinity of the upper and lower membranes was counted. In addition, the NCM method, which was used in previous study, was performed. Therefore, a piece of dental pulp tissue was placed on one cell insert, and the number of cells migrating to the vicinity of the membrane after 120 hours reported in the previous study was counted and compared.” (L103 to L111 on page 3, Revision text).

“3.2. Comparison of the number of cells migrating to the vicinity of the membrane

We compared the number of cells migrating to the vicinity of the membrane between the improved CMDPT of this time and the method in the previous study. As a result, in Improved CMDPT, it was found that the number of cells migrated to the vicinity of the membrane in 48 hours was comparable to the number of cells observed in 120 hours in the previous study (Figure 3). Furthermore, at 72 hours of Improved CMDPT, it was found that the greatly number of cells migrated to the vicinity of the membrane than were observed in previous studies. (Figure 3). From these results, it was found that cells migrate to the vicinity of the membrane early in Improved CMDPT. In addition, it is suggested that DPSCs can be obtained more efficiently in the Improved CMDPT because the moving distance is shorter in the upward and downward direction than in the one direction.” (L187 to L196 on page 5, Revision text)

“Figure 3. Comparison of the number of cells migrating to the vicinity of the membrane

The number of cells migrated to the vicinity of the membrane in 48 hours of Improved CMDPT was comparable to the number of cells observed in 120 hours with NCM. In addition, at 72 hours of Improved CMDPT, it was found that the greatly number of cells migrated to the vicinity of the membrane than had been observed in the previous study.” (L198 to L202 on page 6, Revision text)

  1. Moreover, I recommend that the discussion section be revisited. Currently, it tends to reiterate findings already covered in the results section. Instead, it would be more beneficial if the authors focused on contrasting their new method with traditional DPSC collection techniques and their previously established method.

Response

Thank you for your comment. We added and deleted in discussion as follows:

“Moreover, it was found that the improved method not only migrated cells earlier, but also increased the number of migrated cells compared to the method in the previous study. Compared to the distance required for movement in one direction so far, the total distance cells migrate is reduced because it can move both upward and downward direction. As a result, it is thought that this leads to efficient collection of cells.” (L280 to L285 on page 8-9, Revision text)

L300 to L303 on page 9, Revision text, were deleted.

We believe the manuscript has been improved satisfactorily and hope that it is now acceptable for publication in cells

Yours sincerely,

Reiko Tokuyama-Toda, DDS, PhD 

Round 2

Reviewer 1 Report

Dear authors,

I believe a full characterization of the extracted/isolated cells is needed. Please, for at least three repeats, include the CD markers characterization and ability to differentiate into three lineages.

Thanks

Very minor mistakes

Author Response

Aug 22, 2023

Dear Reviewer #1

cells

Re:   Manuscript ID: cells-2491615

Title: Improved method for dental pulp stem cell preservation and its underlying cell biological mechanism

Thank you for your valuable comments concerning our manuscript entitled " Improved method for dental pulp stem cell preservation and its underlying cell biological mechanism."

We have carefully studied your comments and made the necessary corrections, and are sending here the revised-2 manuscript again. The corrected document has colored text.

Our responses to your comments are as follows:

Response to the comments of Reviewer #1

  1. Dear authors, I believe a full characterization of the extracted/isolated cells is needed. Please, for at least three repeats, include the CD markers characterization and ability to differentiate into three lineages. Thanks

Response

Thank you for your comment. We appreciate your suggestions for improving our manuscript. Following your suggestion, I added a supplemental figure of the DPSCs character collected in this method (supplemental figure 1). Also, along with that, I added a few words in manuscript. (L288 on page 9, Revision-2 text)

I apologize for taking so long time to resubmit.

We believe the manuscript has been improved satisfactorily and hope that it is now acceptable for publication in cells

Yours sincerely,

Reiko Tokuyama-Toda, DDS, PhD 

Round 3

Reviewer 1 Report

I am happy with the modifications.

Very few improvements